# Monovalent metal ion binding promotes the first transesterification reaction in the spliceosome

Jana Aupič [1], Jure Borišek[2], Sebastian M. Fica [3], Wojciech P. Galej [4] & Alessandra Magistrato [1] ✉

Cleavage and formation of phosphodiester bonds in nucleic acids is accomplished by large cellular machineries composed of both protein and RNA. Long thought to rely on a two-metal-ion mechanism for catalysis, structure comparisons revealed many contain highly spatially conserved second-shell monovalent cations, whose precise function remains elusive. A recent high-resolution structure of the spliceosome, essential for pre-mRNA splicing in eukaryotes, revealed a potassium ion in the active site. Here, we employ biased quantum mechanics/ molecular mechanics molecular dynamics to elucidate the function of this monovalent ion in splicing. We discover that the $K^+$ ion regulates the kinetics and thermodynamics of the first splicing step by rigidifying the active site and stabilizing the substrate in the pre- and post-catalytic state via formation of key hydrogen bonds. Our work supports a direct role for the $K^+$ ion during catalysis and provides a mechanistic hypothesis likely shared by other nucleic acid processing enzymes.

Many DNA and RNA processing enzymes rely on the precise positioning of two divalent ($M^{2+}$) metal ions to accomplish their catalytic function[1-4]. This two-metal-ion mechanism is highly conserved and well-studied; both ions act as Lewis acids, with one activating the nucleophile (water or sugar hydroxyl) attacking the scissile phosphate group, while the other stabilizes the leaving group. Conversely, the role of monovalent ions in nucleic acid processing machines has been hitherto largely overlooked and underexplored. In fact, it recently emerged that due to assignment errors, the number of $Mg^{2+}$ ions in these structures has been overestimated[5]. For example, re-analysis of the ribosome structure revealed that $K^+$, rather than $Mg^{2+}$, is the predominant ion in the first solvation shell[5,6]. $K^+$ and $Mg^{2+}$ ions significantly differ in how they interact with DNA and RNA molecules. While $Mg^{2+}$ ions bind strongly to phosphate groups and therefore have an organizing and stabilizing effect, $K^+$ is a chaotropic ion that promotes mobility of coordinating nucleotides[5]. Moreover, while it was initially thought the main function of monovalent ions was to buffer the negative charge associated with the phosphodiester backbone and

thus promote correct folding and assembly of multi-component nucleic acid processing enzymes, emerging evidence suggests they also play a pivotal role in mediating functional conformational changes and the catalytic process[3,7]. Therefore, the correct and complete identification of metal ions is highly relevant for untangling the dynamics, assembly, function, and regulation of nucleic acid processing machines.

A crucial step in the maturation of the precursor of messenger RNA (pre-mRNA) is the excision of introns from pre-mRNA in a process called splicing[8,9]. In eukaryotes, splicing is carried out by the spliceosome, a large ribonucleoprotein machinery that catalyzes two $S_N2$-type transesterification reactions. To accomplish this task, the spliceosome must cycle through several states (E, A, B, $B^{act}$, $B^*$, C, $C^*$, P, ILS) that differ in their composition and conformation[10-12]. Cryo-EM analysis of different spliceosome states has yielded a comprehensive understanding of the splicing process[13-15]. Briefly, the first transesterification reaction, called branching, occurs in the $B^*$ state, while the second exon-ligation step occurs in the $C^*$ state. In the $B^*$ state, pre-mRNA is

[1]National Research Council of Italy (CNR)—Materials Foundry (IOM) c/o International School for Advanced Studies (SISSA), Trieste, Italy. [2]Theory department, National Institute of Chemistry, Ljubljana, Slovenia. [3]Department of Biochemistry, University of Oxford, Oxford, UK. [4]European Molecular Biology Laboratory, Grenoble, France. ✉e-mail: alessandra.magistrato@sissa.it

positioned in the active site of the spliceosome, containing two catalytic $Mg^{2+}$ ions (M1 and M2) by small nuclear RNAs (snRNAs) U2, U5, and U6, held in place by the Prp8 protein and extensive network of protein-protein and protein-RNA interactions (Fig. 1a, b)[16]. The active site is additionally stabilized by branching-specific protein factors Yju2 and Cwc25, which allow stable docking of the U2 snRNA-intron helix (branch helix) to the active site[17,18]. Branching occurs when the 2′-OH of the highly conserved intron adenosine (branch point adenosine, BPA), primed by M2, attacks the scissile phosphate at the 5′-splice site, held in place by both M1 and M2 (Fig. 1b, c). The reaction results in the formation of an intron lariat-3′-exon intermediate and a cleaved 5′-exon[19]. Branching is followed by spliceosome remodeling, during which Yju2 and Cwc25 dissociate from the complex, and the branch helix is displaced from the active site to make room for the 3′-splice site[20]. Subsequently, the 3′-OH at the 3′-end of the 5′-exon carries out a nucleophilic attack on the 3′-splice site, excising the intron in lariat form and producing the mature mRNA[21].

Intriguingly, a recent cryo-EM structure of the spliceosome right after branching (C state), resolved at 2.8 Å, identified an additional, previously unassigned $K^+$ ion coordination site (K1, Fig. 1a), positioned in the direct vicinity of the two catalytic $Mg^{2+}$ ions[22]. Before catalytic activation of the spliceosome, this $K^+$ site is blocked by a protein side chain, consistent with a key role during catalysis. Indeed, ion replacement experiments showed the $K^+$ ion significantly increased forward splicing efficiency. These observations suggest the $K^+$ ion may be more directly involved in the catalytic process than previously thought.

Here, we apply density functional theory (DFT) quantum mechanics (QM)/molecular mechanics (MM) molecular dynamics (MD) simulations to address the mechanistic role of the $K^+$ ion in the branching step of the splicing reaction. We demonstrate that, by rigidifying the binding pocket of the two $Mg^{2+}$ ions and correctly orienting the BPA for nucleophilic attack, the $K^+$ ion takes an active role in the catalytic process, substantially decreasing the activation free energy

barrier and stabilizing the product state, thus promoting the forward splicing reaction. Comparison with group II introns suggests its enzymatic role evolved early and is conserved across other splicing machines.

## Results

### The $Mg^{2+}$/$K^+$ cluster primes the BPA for nucleophilic attack

We based our calculations on the cryo-EM structure of the spliceosome from *Saccharomyces cerevisiae* stalled immediately after branching in the C complex state, which retains the branching splicing factors and precedes Prp16 remodeling[22]. The structure was reconstructed at 2.8 Å resolution for the complex core (Fig. 1a) and the model was also shown to match well the electron density map obtained for the pre-branching B* complex at a lower resolution (4 Å)[16], indicating no active site rearrangement between the two states. Indeed, O3′ at the 3′-end of the 5′-exon (G(−1)), O2′ of the BPA, and *pro*-$R_p$ O from the first residue of the intron region (G(+1)) are all coordinated to catalytic $Mg^{2+}$ ions M1 and M2 in the C complex model ($d_{coord} = 2.0$–2.4 Å). $Mg^{2+}$ ions are additionally coordinated by U6 snRNP residues A59, G60, G78, and U80. The $K^+$ ion displays octahedral coordination geometry with bond distances of 2.7–3.0 Å and coordinating oxygen atoms provided by G52, A59, G60, and U80 (Fig. 1b).

To evaluate the stability of the proposed coordination spheres, we first equilibrated the system by classical MD simulations, followed by more accurate QM/MM MD, based on the DFT-BLYP level of theory for the QM part and on Amber ff14SB (protein) and $\chi_{OL3}$ (RNA) force fields for the MM part (Supplementary Fig. 1)[23]. Since in the C state, the branching reaction is already completed, we rebuilt the reactant state by breaking the non-canonical 2′–5′ bond between BPA and G(+1) and restoring the bond between G(−1)-O3′ and G(+1)-P (Fig. 1b). To reduce computational costs, only central protein and RNA components were used in the simulations (see Methods). Simulations were performed in explicit water, resulting in a system containing 535,391 atoms

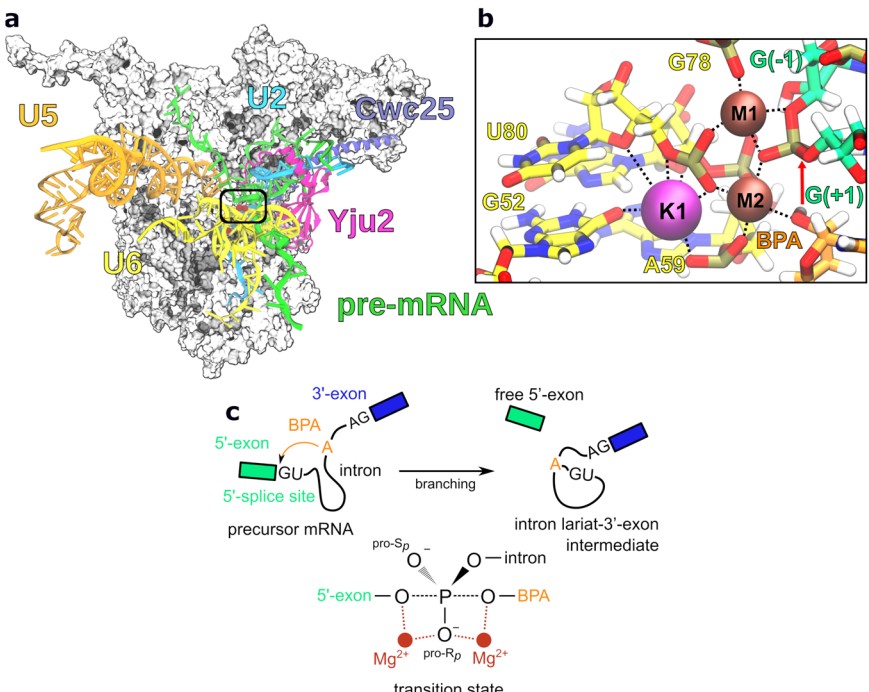

**Fig. 1 | Scheme of the spliceosome complex pre-organized for the first transesterification reaction. a** Cryo-EM structure of the yeast C complex spliceosome (PDBID: 7B9V) used in this study. Relevant splicing factors (Yju2, Cwc25), small nuclear RNAs (U2, U5, and U6), and pre-mRNA are shown in the new cartoon representation. The protein core is shown as a white surface. **b** Close up view of the active site containing two $Mg^{2+}$ ions (M1 and M2) and $K^+$ ion (K1) along with their coordinating residues. **c** Branching is initiated by the attack of the branch point adenosine (BPA) in the intron sequence on the 5′-splice site, leading to the formation of the intron lariat-3′-exon intermediate and free 5′-exon.

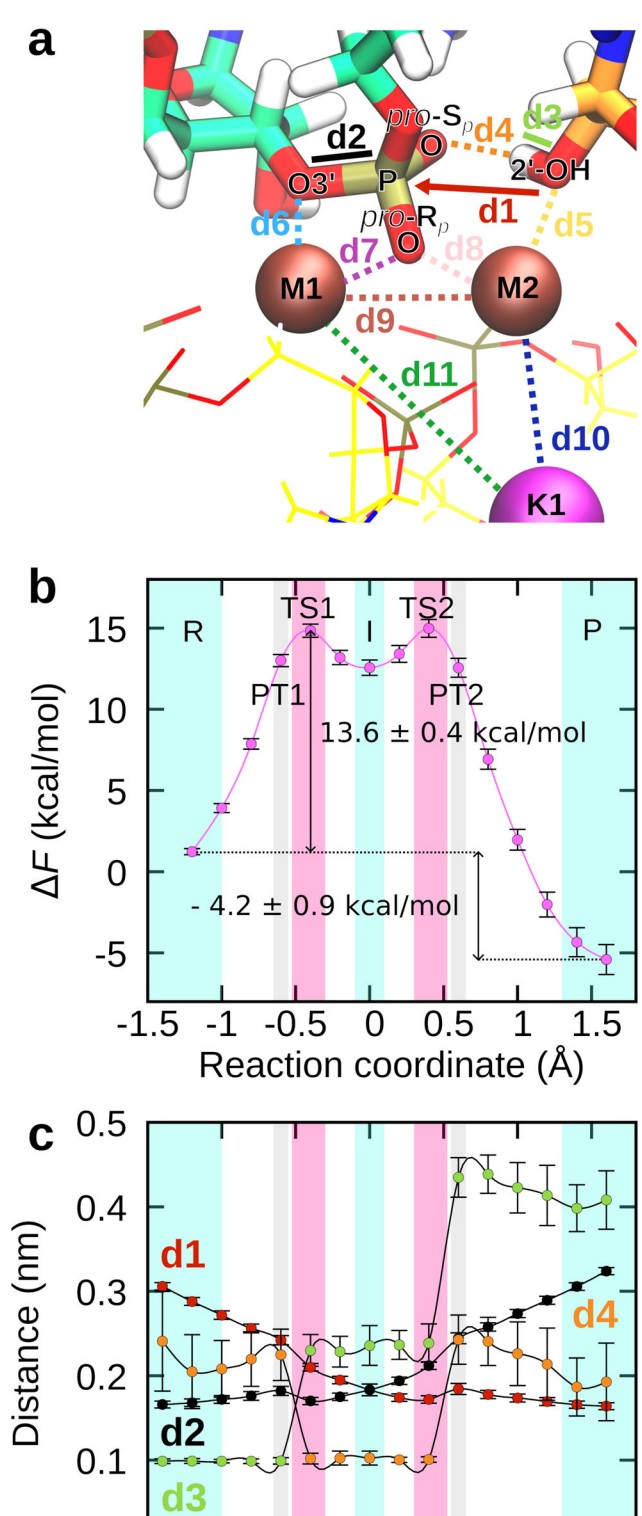

**Fig. 2 | Branching reaction follows a two-step associative mechanism. a** Close-up view of the reaction site in the reactant state. The reaction proceeds by the nucleophilic 2'-OH attacking the scissile phosphate (P), with O3' acting as the leaving group. Distances between relevant atoms are labeled from d1 to d11. $Mg^{2+}$ ions are labeled as M1 and M2, $K^+$ as K1. **b** Helmholtz free energy ($F$) as a function of the reaction coordinate (RC). The reactant (R) and the product state (P) are separated by the phosphorane-like intermediate state (I). The reaction is characterized by two transition states (TS1 and TS2) and two proton transfer steps (PT1 and PT2). The free energy profile was obtained by integrating the mean constraint force along the RC. At each RC, the mean constraint force was obtained over 6000 frames. The corresponding errors were obtained from SD using error propagation. **c** Distances between atom pairs depicted in panel a as a function of the reaction coordinate. For clarity, only distances exhibiting a marked change as the reaction progressed are shown. Data are presented as mean values ± SD ($n$ = 1000 frames). Source data are provided as a Source Data file.

while $K^+$ was located at a distance of 4.3 ± 0.2 Å and 6.2 ± 0.2 Å from M2 and M1, respectively (Supplementary Fig. 3).

## $K^+$ ion lowers the free energy barrier and promotes splicing

To simulate the reaction, we carried out biased QM/MM MD free energy simulations with thermodynamic integration (TI). The reaction was followed along a single reaction coordinate (RC). The latter was defined as the difference of lengths of the breaking bond (G(−1)-O3'–G(+1)-P, d2 in Fig. 2a) and the forming bond (BPA-O2'–G(+1)-P, d1 in Fig. 2a). This protocol was previously applied in the study of splicing in group II introns, where detailed structures of both the pre- and post-catalytic states are available[24,25]. RC was sampled at 16 different points, spanning the range from −1.4 to 1.6 Å (Fig. 2b). The obtained free energy profile (Fig. 2b) exhibited two transition states (TS) separated by a shallow intermediate (I) state. The Helmholtz activation free energy barrier to reach the I state from the reactant (R) state was significantly larger ($\Delta F^{\ddagger}$ = 13.6 ± 0.4 kcal/mol) than that required to arrive from the I to the product (P) state ($\Delta F^{\ddagger}$ = ~ 2 kcal/mol), making the first transition the rate-limiting step of the branching reaction. The reaction free energy was negative ($\Delta F$ = −4.2 ± 0.9 kcal/mol), indicating intron lariat formation is energetically favorable. In comparison, previous static QM/MM calculations performed on a spliceosome model lacking the $K^+$ ion estimated $\Delta F^{\ddagger}$ and $\Delta F$ at 21.5 kcal/mol and -1.8 kcal/mol, respectively[26]. This suggests the $K^+$ ion importantly lowers the activation-free energy barrier for branching and stabilizes the product state, thus promoting forward splicing.

To pinpoint how the reaction proceeds from the reactant to the product state, we monitored key distances between relevant atoms in the active site (Fig. 2a, c, Supplementary Fig. 4, Supplementary Movie 1). Before TS1 (R in Fig. 3), G(−1)-O3' was stably coordinated to M1 (d6), while the attacking 2'-OH group was bound to M2. G(+1)-O($R_p$) interacted with both $Mg^{2+}$ ions. As 2'-OH was approaching the scissile phosphate group a stable hydrogen bond was observed between the 2'-OH proton and G(+1)-O($S_p$) (d4, hb1 in Fig. 3). As the reaction approached the rate-limiting TS1 the proton was transferred from BPA-O2' to G(+1)-O($S_p$), thus aiding BPA-O2'–G(+1)-P bond formation. Proton transfer (PT1 in Fig. 3) was observed at BPA-O2'–G(+1)-P distance 2.42 ± 0.05 Å. The importance of proton transfer to G(+1)-O($S_p$) for the rate of the splicing reaction is in accordance with sulfur substitution experiments[19]. In the intermediate state (I in Fig. 3), both G(−1)-O3' and BPA-O2' were bound to the G(+1)-P, with the forming and breaking bond at approximately the same length (1.8 ± 0.1 Å). The observed phosphorane intermediate indicates the reaction followed an associative mechanism as predicted by Steitz and Steitz for two-$Mg^{2+}$-ion aided catalysis[1]. This is in contrast to the previous static QM/MM study, where the reaction was observed to follow a dissociative mechanism and no reaction intermediate was observed[26]. Here, the intermediate state was stabilized by both $Mg^{2+}$ ions with G(−1)-O3' and BPA-O2'

(Supplementary Table 1). After 5 ps of QM/MM MD, the coordination spheres of M1, M2, and K1, as observed in the cryo-EM structure[22], were still stable, with average coordination distances from 2.0 to 2.3 Å for $Mg^{2+}$ ions and 2.8 to 3.4 Å for the $K^+$ ion (Supplementary Fig. 2). The nucleophilic 2'-OH group from BPA was optimally positioned for attacking the scissile phosphate group with BPA-O2'–G(+1)-P distance of 3.3 ± 0.2 Å. The catalytic $Mg^{2+}$ ions were positioned 4.1 ± 0.1 Å apart,

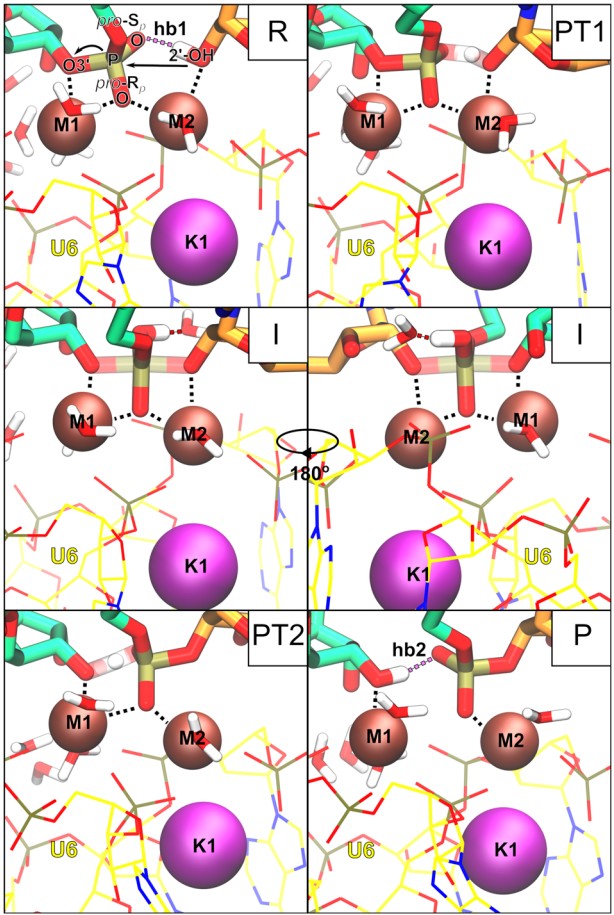

**Fig. 3 | Hydrogen bonding facilitates proton transfer from the nucleophile to the leaving group via the scissile phosphate.** The active site is shown in the reactant state (R), during the first proton transfer (PT1), in the intermediate state (I), during the second proton transfer (PT2), and in the product state (P). The intermediate state is stabilized by water-assisted hydrogen bonding.

positioned at a distance of $2.2 \pm 0.1$ Å and $2.4 \pm 0.2$ Å from M1 and M2, respectively, and G(+1)-O($R_p$) approximately 2.2 Å away from both M1 and M2. Additionally, we observed hydrogen bonding of the proton bound to G(+1)-O($S_p$) to a surrounding water molecule that approached the active site from the bulk solvent treated at the MM level. Upon interaction with the active site, the water molecule was added to the QM zone. As G(−1)-O3' moved away from G(+1)-P, breaking the 5'-exon-intron link and overcoming an almost barrier-less TS, the proton from G(+1)-O($S_p$) (PT2 in Fig. 3) rotated away from the water molecule, breaking the hydrogen bond, and passed to G(−1)-O3'. Moreover, with the reaction approaching the P state (P in Fig. 3), M1–G(−1)-O3' coordination bond length decreased (Supplementary Figs. 4 and 5), additionally stabilizing the negative charge of the leaving group. Conversely, G(+1)-O($R_p$) and BPA-O2' detached from M1 and M2, respectively (Supplementary Figs. 4 and 5). Additionally, the distance between M1–M2 slightly increased in the product state ($4.4 \pm 0.2$ Å). During the reaction, K+ was stably coordinated at its binding site, with larger fluctuations in coordination bond length observed only for G60 (d5, Supplementary Fig. 5), and at a constant distance from both M1 and M2.

## Li+ ion at the K1 site reduces the rate and favourability of branching

Experiments analyzing branching efficiency in the presence of different monovalent ions indicated that Li+ ions promote reversal of the reaction, thus decreasing splicing efficiency[22]. To check whether our simulations could replicate and explain the observed differences, we substituted the K+ ion with Li+ and repeated the calculations. First, we applied unbiased QM/MM MD simulation to assess the stability of the modified active site. Our simulations revealed the Li+ ion was not able to form a stable coordination sphere at the K1 site (Supplementary Fig. 6), reflected in altered coordination bond lengths and larger fluctuations. More specifically, Li+ was observed to shuttle between two different binding poses (Supplementary Fig. 7). In Binding pose 1, Li+ coordinated to G52 and U80 (sugar oxygen and O3'), while in Binding pose 2 Li+ binds to A59, G60 and U80 (O3' and pro-$R_p$ O). This, in turn, resulted in large fluctuations in M1–Li+ and M2–Li+ distances (Supplementary Fig. 8). In both cases, water molecules are transiently bound to the Li+ ion to complete the coordination sphere. The observed binding flexibility is not surprising since Li+ and K+ ion size and coordination preferences differ significantly. While Li+ prefers a tetrahedral coordination geometry with coordination bond lengths of ~2 Å[27], the K+ coordination sphere is at least octahedral with coordinating atoms lying ~3 Å away from the metal center[28].

Nevertheless, the coordination spheres of Mg²⁺ ions M1 and M2 were not significantly affected. While 2'-OH of BPA was less stably coordinated to M2 ($2.9 \pm 0.5$ Å), it was still well positioned to carry out the nucleophilic attack ($3.4 \pm 0.2$ Å, Fig. 4a and Supplementary Fig. 6). However, biased QM/MM MD simulations coupled with TI revealed a markedly different reaction profile (Fig. 4b) with the activation free energy barrier for the rate-limiting step increasing to $16.2 \pm 0.5$ kcal/mol. Moreover, albeit the reaction was still exergonic, the $\Delta F$ reduced to $-2.3 \pm 0.9$ kcal/mol, driven by reduced stability of the product state, in accordance with experimental observations showing increased reversal of the reaction[22].

To home in on the underlying cause for the observed differences, we compared the reaction process to that observed for K+ ions (Fig. 4c, Supplementary Figs. 9 and 10). On the whole, the branching reaction progressed similarly as in the presence of K+ ions with two TS associated with bond formation and bond breaking, water-stabilized phosphorane intermediate, and a proton transfer from BPA-O2' to G(−1)-O3' via G(+1)-O($S_p$) (Fig. 5a). Distances between atom pairs in the active site followed the same trend as when K+ was bound at the K1 site, with exception of M1–Li+ and M2–Li+ distances, which showed large fluctuations, as observed in unbiased QM/MM MD (Supplementary Fig. 10).

Studies on metalloenzymes have demonstrated that second shell cations can greatly affect reactivity by modulating local electronic properties, such as local charge distribution, electric dipole moments, and electric field[29–31]. Thus, we speculated the observed changes in the reaction profile might be due to differences in charge distribution, affecting the nucleophilic character of the attacking hydroxyl group or the stability of the leaving group. Partial atom charges along the RC were calculated with the D-RESP method, however replacing K+ with Li+ did not have a significant effect on partial atomic charge distribution (Supplementary Fig. 11). Additionally, we compared how K+ and Li+ ions influence the local electric field in the active site in the rate-limiting TS1 (Supplementary Fig. 12)[29]. The analysis revealed that in the presence of K+, the forces acting on reactive atoms optimally promoted splicing. The electric field pointed the 2'-OH of BPA towards the scissile phosphate group, while the leaving G(−1)-O3' was directed away from G(+1)-P. In the presence of Li+, the electric field was still in a configuration aligned with the splicing reaction; however, the directionality of the forces was less ideal. By contrast, the relative mean square fluctuation of individual residues involved in the reaction or active site stabilization, most prominently BPA, G(+1), A59, and U80, were somewhat higher when the reaction was simulated in the presence of Li+ (Supplementary Fig. 13), indicating a less rigid catalytic scaffold. The comparison of distances between relevant atoms in the active site revealed further subtle differences (Fig. 4a, c and Supplementary

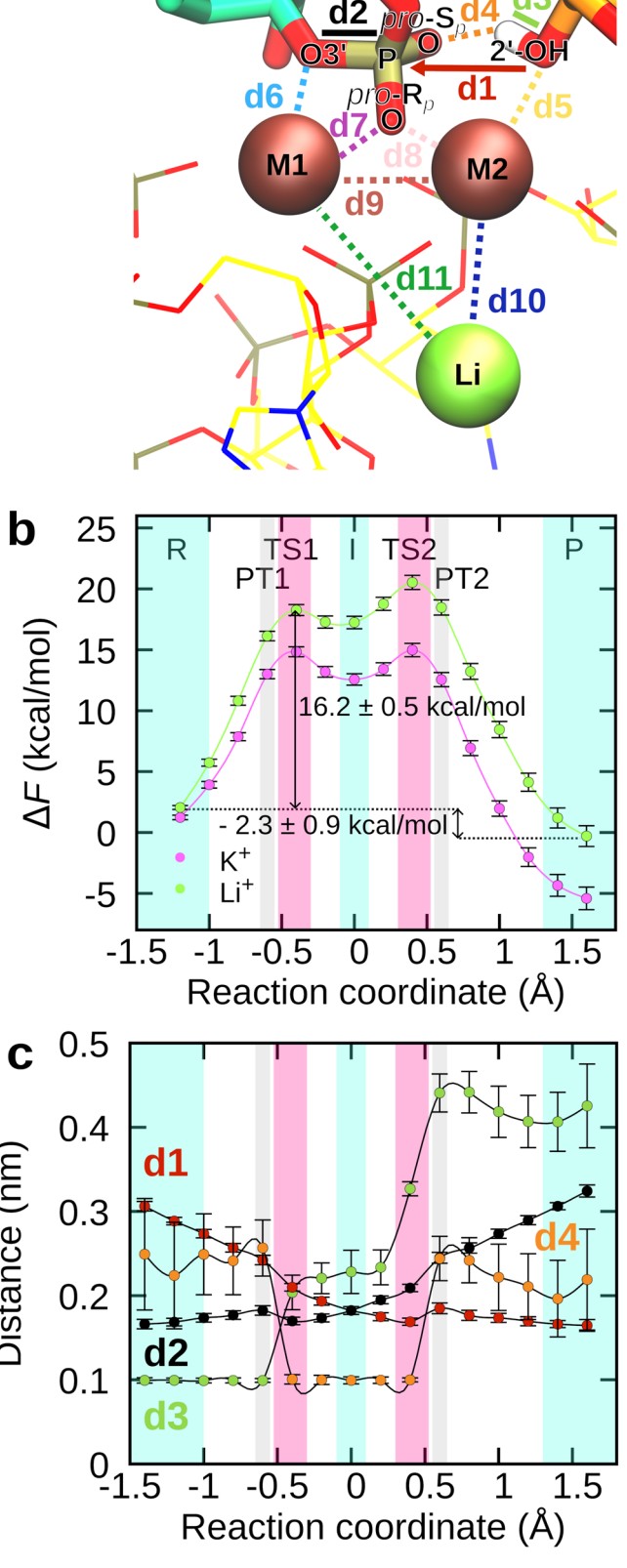

**Fig. 4 | Substitution of K⁺ ion by Li⁺ increases the activation barrier for the branching reaction. a** Active site in the pre-reaction state after equilibration with QM/MM MD. **b** Helmholtz free energy (*F*) as a function of the reaction coordinate (RC) in the presence of Li⁺ ion (in green) in comparison to the free energy profile obtained when K⁺ ion was bound in the active site instead (in violet). As in the case of the K⁺ ion, the reaction exhibits an intermediate state (I), two transition states (TS1 and TS2), and two proton transfer steps (PT1 and PT2). The arrows denote the activation barrier and the free energy difference between the reactant (R) and product state (P). The free energy profiles were obtained by integrating the mean constraint force over the RC. At each RC value, the mean constraint force was calculated from the last 6000 frames. The corresponding errors were obtained from SD using error propagation. **c** Distances between atom pairs depicted in panel a as a function of the reaction coordinate. Data are presented as mean values ± SD (*n* = 1000 frames). For clarity, only distances exhibiting a marked change during the reaction are shown. Source data are provided as a Source Data file.

O(S$_p$) (d4 in Fig. 4a, hb1 in Fig. 5a) was larger and displayed higher fluctuations than when K⁺ was present in the active site, indicating a less stable hydrogen bond. Disruption of hydrogen bonding drives up the energy of the system and could explain the observed differences in the reaction profiles. Moreover, this hydrogen bond (hb1) preceded and facilitated the proton transfer from the nucleophilic 2′-OH of BPA to the scissile phosphate, crucial for BPA-O2′–G(+1)-P bond formation. In light of this observation, we analyzed the frequency of hydrogen bonding between 2′-OH of BPA and G(+1)·O(S$_p$) (hb1 in Fig. 5a) before PT1 as the reaction progressed from R to TS1 (−1.4 Å < RC < −0.6 Å), and of the hydrogen bonding between 3′-OH of G(−1) and G(+1)·O(S$_p$) (hb2 in Fig. 5a) after PT2 as the reaction approached the product state. We considered the hydrogen bond formed if the distance between the proton and hydrogen bond acceptor was below 2.2 Å and the angle between the proton donor, transferred proton, and proton acceptor did not significantly deviate from linearity (180 ± 50°, Supplementary Figs. 15 and 16)[32]. Strikingly, for both hb1 and hb2, hydrogen bonding was more commonly observed when K⁺ was bound at the monovalent ion binding site (Fig. 5b), consistent with both lower Δ*F*‡ and Δ*F*. We posit the observed differences are most likely attributable to the unstable binding of the Li⁺ ion, which increases the flexibility of its monovalent ion binding site. As this site shares 3 ligating U6 snRNA residues with the catalytic M2 ion (A59, G60, and U80), this perturbation is further transferred to the M2-coordinated BPA, culminating in reduced hydrogen bonding, impeding proton transfer and consequently splicing efficiency. Additionally, the intrusion of a larger number of water molecules into the monovalent ion binding site likely further contributed to the distortion of the catalytic site (Supplementary Fig. 17).

## Discussion

The expansion of the two-metal-ion catalytic motif by second-shell cations has been reported for various DNA and RNA processing enzymes[2,3,33–35]. While the first identified examples contained only divalent second-shell cations, recently monovalent ions were also found at equivalent positions[3,36,37]. Due to their conserved spatial localization, a shared role in the enzymatic reaction mechanism was proposed[36,37]. Classical MD indicated monovalent cations near the active site modulate substrate docking and product release;[38] however, their importance for the chemical reaction itself, i.e., phosphodiester bond breaking and formation, remained unaddressed. QM/MM MD simulations, while very demanding computationally, particularly for a large system such as the spliceosome complex, offer a unique opportunity to examine reaction mechanisms at an atomic level of detail and analyze how individual active site components affect reaction thermodynamics.

Our simulations of the spliceosome active site revealed for the first time that the K⁺ ion is directly involved in the catalysis of the

Fig. 10). In TS2, the hydroxyl group of the scissile phosphate formed a hydrogen bond with G(−1)-O2′ instead of a water molecule, as observed in simulations with K⁺ in the active site, (Supplementary Fig. 14), explaining the somewhat larger second activation free energy barrier (-3.2 kcal/mol). More importantly, as the reaction approached TS1, the average distance between 2′-OH of BPA proton and G(+1)-

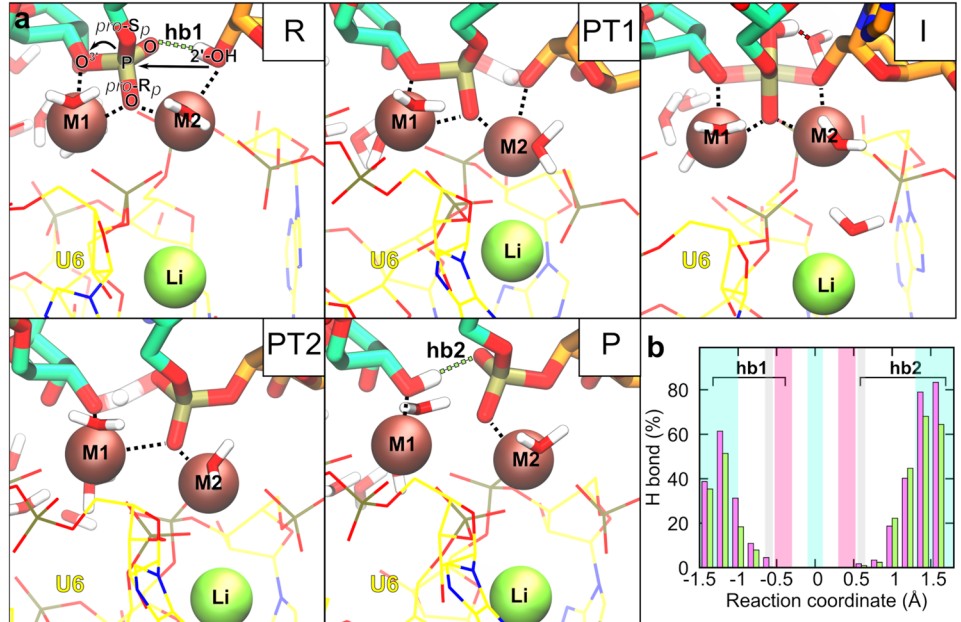

**Fig. 5 | Li⁺ ion negatively affects hydrogen bonding in the active site and hinders proton transfer. a** Snapshots of the active site during the branching reaction simulated in the presence of the Li⁺ ion. The active site is shown in the reactant state (R), during the first proton transfer (PT1), in the intermediate state (I), during the second proton transfer (PT2), and in the product state (P). **b** Frequency of hydrogen bonding between 2'-OH of BPA and G(+1)-O($S_p$) before the branching reaction (hb1) and 3'-OH of G(−1) and G(+1)-O($S_p$) after the reaction (hb2) as a function of the reaction coordinate. The violet and green bars report values obtained when K⁺ or Li⁺ ion, respectively, was bound in the active site. The distributions of distances and angles between relevant atoms, which served as the basis for calculating the frequency of hydrogen bonding, are shown in Supplementary Figs. 15 and 16. Source data are provided as a Source Data file.

branching reaction. K⁺ stabilized the active site in both the pre- and post-branching state by facilitating optimal positioning of reactive moieties. This resulted in both a lower activation-free energy barrier and reaction-free energy as compared to those obtained when the reaction was previously investigated in the absence of the monovalent ion[26]. However, the previous study relied on static QM/MM calculations and a starting spliceosome structure that required remodeling of the position of Mg²⁺ ions in the active site, which might have also partially contributed to the observed differences in the free energy profile. The stabilizing effect of the monovalent ion was diminished when the K⁺ ion was replaced with Li⁺, in line with experimentally determined splicing efficiency in different ionic buffers[22]. Thus, our work confirms that the K⁺-induced boost in splicing efficiency is not due to unspecific modulation of the electrostatic environment induced by the binding of a generic monovalent ion in the vicinity of the two-Mg²⁺-ion motif but is rather brought about by the precise positioning of the K⁺ ion in the heteronuclear metal cluster. In contrast, Li⁺ could not seamlessly substitute the K⁺ ion since the two ions significantly differ in their size and charge densities and, thus, in their RNA coordination preferences[39]. In principle, the experimentally observed reduction in splicing levels in the presence of Li⁺ ions may additionally be due to large-scale structural transitions, moving the spliceosome away from the branching conformation. However, these phenomena occur on microsecond to millisecond timescales, not accessible to the QM/MM MD simulations performed here and would require the use of enhanced sampling techniques in combination with classical force fields, which inadequately describe divalent metal ions, leading to less reliable simulation outcomes. Nonetheless, arguing against large conformational changes, purified spliceosomes could be switched easily between buffers containing several different monovalent ions without losing catalytic competence[22].

Our results demonstrate that second-shell monovalent ions can be active participants in the catalytic process, just as their divalent counterparts. This suggests the chemistry of nucleic acid processing enzymes is more varied than previously believed and not reliant solely on divalent metal ions. The heterogeneous nature of the metal cluster might serve to add another regulatory layer to the catalytic process with binding/unbinding events regulating conversion rates in a more complex manner.

Comparison with group II introns, from which the spliceosome and its catalytic mechanism evolved, gives further credence to the generality of the mechanistic role of monovalent ions proposed here. Group II introns contain two additional K⁺ ions in the vicinity of the two-Mg²⁺-ion site, with one (K1) perfectly overlapping with the K⁺ ion position in the spliceosome (PDBID: 4FAR, Supplementary Fig. 18)[37]. Indeed, structural studies have shown that K1 rigidifies the architecture of the divalent ion binding site[37]. Additionally, its binding/unbinding, propelled by protonation of C358 (A59 in the yeast spliceosome), enabled conformational changes needed for splicing to progress from branching to the exon-ligation step[36]. In light of similarities between group II introns and the spliceosome, it is tempting to speculate that in addition to promoting branching, K⁺ ion binding might also facilitate stabilization of the active site following large-scale conformational changes. Intriguingly, analysis of cryo-EM maps suggested K⁺ ion binding is coupled to BPA docking at the active site, indicating K⁺ ion binding and unbinding may indeed be linked to transitions between different spliceosome states[22]. Additionally, since K⁺ is also in the active site during the second splicing step, it might also promote exon ligation[20,22].

In summary, while K⁺ ions were first recorded as an essential cofactor in splicing in 1984[40], it took almost 40 years to definitively localize its binding site, allowing us to assess its role in the catalytic mechanism precisely. We demonstrate that the monovalent ion (K⁺) delicately and specifically tunes the active site geometry to promote optimal hydrogen bonding, vital for both guiding the nucleophilic attack of the BPA and stabilizing the product state, and is thus essential for regulating the splicing reaction. In conclusion, our findings identify monovalent metal ions as important regulators of RNA processing machines and supply a mechanistic hypothesis that may be broadly applicable to other RNA processing complexes.

## Methods

### System setup and classical MD simulations

The starting structure was taken from the electron cryo-microscopy reconstruction of the *Saccharomyces cerevisiae* spliceosome in the C complex state (PDBID: 7B9V)[22]. To reduce the system size and the associated computational cost, only protein domains and RNA molecules relevant for the first reaction step were preserved, while others were excluded from the calculations. The final complex thus contained Prp8, Yju2, CLF1 isoform 1, snRNPs U2, U5, and U6, and relevant pre-mRNA fragments. While splicing factor Cwc25 is also essential for locking the spliceosome in the branching conformation, it makes limited contact with the active site[22,41]. Preliminary simulations for the spliceosome system lacking Cwc25, showed the removal of Cwc25 had no negative effect on the structure of the active site (Supplementary Fig. 2, Supplementary Table 1). Conversely, removing Yju2 resulted in less stable coordination of BPA-O2' to M2 ($3.2 \pm 0.7$ Å) and lengthening of the BPA-O2'-G(+1)-P distance ($3.6 \pm 0.3$). Therefore, we studied the reaction mechanism of the branching reaction in the absence of Cwc25 while Yju2 was retained in the system.

The active site was solvated with the SOLVATE program (version 1.0)[42], which is able to solvate poorly accessible cavities. Next, the spliceosome complex was placed in a cubic box (177 Å × 180 Å × 168 Å) and further solvated with TIP3P water using the TLEAP tool from the AMBERTOOLS20 package[43]. $Na^+$ ions were added to the simulation box to achieve electroneutrality. Protonation states of ionizable residues were determined by the H++ web tool (version 3.0) at pH 6.5[44] (Supplementary Data 1). The simulated system contained 535,391 atoms in total. Forces in the system were described with Amber force fields ff14SB (protein) and $\chi_{OL3}$ (RNA)[45,46] as recommended by benchmark studies[47]. Metal ions were treated as charged van der Waals spheres. Ions were described with Li and Merz 12 − 6 ion parameters[48,49]. To relax the system and prepare it for QM/MM MD, we first ran classical MD in GROMACS (version 2020.3)[50]. The simulated system was first equilibrated in the NVT ensemble for 10 ns using periodic boundary conditions and position restraints on the protein and RNA components as well as on $Mg^{2+}$ and $K^+$ ions. Next, equilibration was continued in the NPT ensemble for another 15 ns. While position restraints were removed, harmonic bonds ($k = 4627.5$ kJ/mol) were placed between $Mg^{2+}$ and $K^+$ ions and their coordinating atoms to prevent active site distortion due to well-known force field issues when simulating divalent metal ions. Newton's equations of motion were solved with the leap-frog algorithm using a 2 fs time step. Electrostatic interactions were evaluated using the particle mesh Ewald method[51]. The van der Waals and electrostatic cut-off was set at 1.2 nm. Temperature ($T$) was controlled using the V-rescale thermostat ($T = 300$ K, coupling constant was 0.1 ps)[52], while pressure was kept at 1 bar with the Parrinello–Rahman barostat (time constant for pressure coupling was 2 ps)[53].

### Quantum mechanical (QM)/MM MD simulations

QM/MM MD simulations were performed with cp2k (version 9.1)[54]. The QM portion (Supplementary Fig. 1) was composed of the two catalytically active $Mg^{2+}$ ions, the nearby potassium $K^+$ ion, scissile phosphate group, branch point adenosine and nucleotide residues from U6 snRNA that act as ligands for $Mg^{2+}$ ions and the K1 site (G52, A59, G60, G78, U80). Additionally, water molecules in the coordination sphere of $Mg^{2+}$ (2 molecules for M1 and 1 for M2), $K^+$ (1–3 molecules), and $Li^+$ ions (2–5 molecules) were also treated at the QM level. While water molecules were stably coordinated to M1 and M2, exchange with the bulk solvent was observed for $K^+$ and $Li^+$, therefore the list of water molecules in the QM zone was continuously updated. Total number of QM atoms was thus 107 − 119. The quantum box was described with the Becke−Lee−Yang−Parr DFT (DFT/BLYP)[55,56]. Double-ζ molecularly optimized basis set was used along with Goedecker−Teter−Hutter pseudopotentials (DZVP-MOLOPT-GTH)[57]. For the $Mg^{2+}$ and $K^+$ ions, the short-range version of the basis set was used (DZVP-MOLOPT-SR-

GTH-q2, DZVP-MOLOPT-SR-GTH, respectively). The plane wave cutoff was set at 400 Ry, and DFT-D3 dispersion correction was applied[58]. First, the system was relaxed in a microcanonical (NVE) ensemble, followed by a short simulated annealing using the Langevin thermostat. Next, the system was heated in a step-wise manner from 150 K to 300 K using the NVT ensemble. Temperature was maintained with the V-rescale thermostat[52]. Two thermostats were applied; one for the QM region and one for the rest of the system. The time constant for both was 10 fs. Following system equilibration, the trajectory was collected over 5 ps. The time step was 0.5 fs. During production runs, the thermostat time constant was increased to 1000 fs. Cpptraj from AMBERTOOLS20 and python (version 2.7.18) packages MDtraj (version 1.9.3) and MDAnalysis (version 0.20.1) were used for trajectory analysis[59–62]. Dynamical RESP (D-RESP) charges were calculated with cp2k (version 9.1)[63]. For each RC, D-RESP charges were obtained by averaging over 100 frames extracted from the trajectory. The electric field forces acting on reacting atoms were obtained with APBS software (version 1.5)[64] by solving the Poisson-Boltzmann equation. The calculations were performed on the averaged structure of the system in the first transition state (RC = −0.4) using only charges from the spliceosome, excluding neutralizing $Na^+$ ions.

### Free energy calculations

Free energy profiles were obtained by performing blue moon ensemble calculations using a single reaction coordinate (RC)[65–67]. Here, RC was defined as the difference in length of the breaking bond (G(−1)-O3'-G(+1)-P) and forming bond (BPA-O2'-G(+1)-P) as applied for similar systems previously[24]. Starting from the intermediate state (RC = 0 Å), the reaction was sampled in parallel towards the reactant (RC = −1.4 Å) and product state (RC = 1.6 Å). The RC was decreased or increased sequentially in 0.2 Å steps with a growth rate of 0.003 Å/fs. At each RC, the system was simulated for 5 ps, except at RC values −0.6 and 0.6, where proton transfers took place and simulation time was extended to 7 ps to obtain better statistics. This resulted in a total QM/MM MD simulation time of ~80 ps. The mean force required to constrain the atoms at a selected value of the RC was calculated by averaging the Lagrange multiplier of the Shake algorithm after approximately 2 ps of simulation time (4 ps at RC = −0.6 and RC = 0.6) when it reached convergence (Supplementary Fig. 19). Formally, the use of the difference of two distances as an RC requires the inclusion of a correction term when calculating the mean force in order to account for the conformational bias introduced by the constraint (see Supplementary Discussion). However, for our system, the correction term amounted to less than 1% of the mean force and was, as such, significantly lower than the calculated error due to statistical uncertainty. Therefore, it was not included in the free energy profile. The free energy profiles were obtained by integrating the constraint force along the RC with the trapezoid method. The standard error for each simulated window was estimated by error propagation analysis from the standard deviation of the constraint force. The overall errors in the activation barrier and reaction-free energy were similarly calculated from the standard error obtained for each window. To obtain the reaction profile in the presence of $Li^+$ ions, biased QM/MM MD simulations were repeated. As the starting frame at each RC, we used the final frame of the corresponding simulation performed with $K^+$.

### Reporting summary

Further information on research design is available in the Nature Portfolio Reporting Summary linked to this article.

## Data availability

All data in support of the findings of this study are available within the article and in the Supplementary Information. Initial and final configurations of simulated systems have been deposited in Zenodo[62]. Due to their large size, MD simulation trajectories are available from the

corresponding author upon request (alessandra.magistrato@sissa.it). The experimental structures of the spliceosome from *Saccharomyces cerevisiae* in the C complex state and group II intron from *Oceanobacillus iheyensis* are available in Protein Data Bank under accession codes 7B9V and 4FAR, respectively. Source data are provided in this paper.

## Code availability

GROMACS (https://www.gromacs.org/) is a freely available open-source software for running chemical simulations. Cp2k (https://www.cp2k.org/) is a quantum chemistry and solid-state physics software package that is freely available under the GPL license. The AMBERTOOLS20 package (https://ambermd.org/AmberTools.php) and the SOLVATE program (https://www.mpinat.mpg.de/grubmueller/solvate) are likewise available free of charge. Cp2k inputs for running QM/MM MD simulations and custom Python scripts employed for the analysis of collected molecular dynamics trajectories have been deposited in Zenodo[62].

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

## Acknowledgements

J.A. acknowledges financial support from the "Giovanni Fraviga" Italian Association for Cancer Research (AIRC) fellowship. J.B. is supported by the Slovenian Research Agency (P1-0017 and J1-3019). S.M.F. is a Wellcome Trust and Royal Society Sir Henry Dale Fellow (grant number 220212/Z/20/Z). W.P.G. is funded by the European Research Council (ERC) under the European Union's Horizon 2020 research and innovation program (grant agreement No. 950278). A.M. thanks the Italian Association for Cancer Research (project AIRC IG 24514).

## Author contributions

J.A. performed the computer simulations. J.A. and J.B. carried out trajectory analysis. A.M. conceived the project and directed the study with input from S.M.F. and W.P.G. J.A., J.B. and A.M. wrote the initial paper. S.M.F, W.P.G., and A.M. revised the paper. All authors discussed the results and contributed to the paper.

## Competing interests

The authors declare no competing interests.
