## [Peer Review File · Nature Communications]

Monovalent metal ion binding promotes the first transesterification reaction in the spliceosomeReviewer #1 (Remarks to the Author):

In this work a computational study is presented on the first transesterification reaction in the spliceosome, which is an important reaction in biology. The authors noticed several K(+) ions in the crystal structure coordinates and investigated how these influence the kinetics and thermodynamics of the reaction. They observe that these K(+) ions are non-innocent and significantly lower the kinetics of the reaction. This is an important and interesting study that fits the remit of Nature Communications well.

1. In several recent studies the de Visser group also highlighted the effect of ions, such as K(+), in the second coordination sphere on catalysis and redox properties. I would cite those studies here.
2. Are there differences in charge distributions in the transition state that can explain the effect of K(+)?
3. Does the K(+) induce an electric field effect that assist in the reaction? Again some links to previous work may be given.
4. Methods: Total number of atoms in the model.
5. Methods: Give protonation states of ionizable residues.
6. Have calculations without K(+) been performed?

Reviewer #2 (Remarks to the Author):

Aupic et al. present molecular dynamics simulations of the active site of the spliceosome in the presence of the two catalytic magnesium ions as well as a highly conserved potassium ion. This potassium ion is found in the active sites of both the spliceosome and the evolutionarily related group II intron. It was hypothesized that this potassium ion played an important role in catalyzing RNA splicing, but the details were not clear. Through their molecular dynamics simulations, Aupic et al. provide a detailed model for how this potassium ion is essential for efficient RNA splicing. They found that the potassium ion plays a key role in modulating catalysis and is responsible for positioning the splice site substrates within the active site. This work is very significant in that it provides new detailed insight into the mechanism of RNA splicing at the atomic level and explains the conservation of active site metal ion architecture in both the group II intron and the spliceosome. I highly recommend publication of this manuscript in Nature Communications in its current form with no revisions required.

Reviewer #3 (Remarks to the Author):

Aupic and coworkers present an interesting study unraveling the overlooked contribution of monovalent ions, namely potassium, on the first transesterification reaction of the spliceosome. Although the work is completely computational, the results of the simulations are compared with experimental data. Therefore, the manuscript should be interesting for both computational and experimental readers of Nature Communications.

The manuscript is well written and the conclusions are nicely supported by the results of the simulations, as well as by comparison with other nucleic acid processing enzymes. Nonetheless, the authors should clarify some points and add some further analyses before the manuscript can be accepted for publication, in my opinion. Please see below a detailed list of comments.

- Page 6, lines 114-118: Can the authors show the time evolution of the distance between the two Mg ions, as well as the distance between the closest Mg ion and K? How do these distances evolve (and potentially change) during the reaction?

- Page 7, lines 134-136: Can the authors provide further details about the previous static calculations with which they compare? Could it be that the difference in free energy barrier and exothermicity of the reaction is due not only to the lack of K in the previous study, but also to the different level of theory used/neglect of system dynamics/other factors to a lower extent?

- Page 9, lines 167-170: Can the authors clarify whether the water molecule stabilizing the

phosphorane intermediate is treated at the MM or the QM level? As far as I can tell from Figure 3I, this water molecule is not part of the coordination sphere of Mg ion, is my understanding correct?

- Page 11: As far as I understand from the Methods section, the initial snapshots for the QM/MM simulations with Li were taken from the K simulations, replacing one monovalent ion by the other. If my understanding is correct, can the authors speculate whether further sampling of the conformational flexibility with Li using classical MD simulations could reveal further structural changes? In addition (and out of curiosity), I was wondering if there is experimental data for the substitution of K by other monovalent ions, e.g. Na.

- Page 17, line 336: I believe there is a typo in the sentence "While splicing factors Cwd25 is also essential [...]"

- Page 17, second paragraph: Can the authors indicate the total number of atoms of the simulated system? Is there any rationale why was the system neutralized with Na ions instead of K? Do the authors expect any difference using Na or K? or no other monovalent ions bind to the spliceosome structure during the simulations? The authors mentioned they use distance restraints between Mg ions and their coordinating atoms during the classical MD to avoid the known issues of classical force fields describing divalent ions, but why such distance restraints were also needed for K and its coordinating atoms? Did the authors want to avoid some ion/water exchange with the solution?

- Page 18, QM/MM section: Can the authors clarify if there are water molecules in the first coordination sphere of any of the two Mg ions that might have been treated at the QM level? How many QM atoms are in total? I believe it would be informative to include in the SI a figure showing the QM region, as well as indicating the link atoms or capping hydrogens used to saturate the QM region, if any.

- Page 18, Free energy section: Can the authors spell out if the different simulation time for each RC (i.e. between 5-7 ps) depends on how fast the constraint force for the corresponding window converges? Also, as far as I remember, in the blue moon ensemble formalism by Ciccotti and coworkers there is a correction to be added that depends on the type of RC used. In particular, such correction should be zero for a simple distance but different from zero for a difference of distances. If my memory serves, have the authors considered such correction in their free energy profile?

RESPONSE TO REVIEWER COMMENTS

Reviewer #1 (Remarks to the Author):

In this work a computational study is presented on the first transesterification reaction in the spliceosome, which is an important reaction in biology. The authors noticed several K(+) ions in the crystal structure coordinates and investigated how these influence the kinetics and thermodynamics of the reaction. They observe that these K(+) ions are non-innocent and significantly lower the kinetics of the reaction. This is an important and interesting study that fits the remit of Nature Communications well.

1. In several recent studies the de Visser group also highlighted the effect of ions, such as K(+), in the second coordination sphere on catalysis and redox properties. I would cite those studies here.

We thank the reviewer for drawing attention to these relevant works. We have included a short comment and added citations to most relevant publications on page 13.

2. Are there differences in charge distributions in the transition state that can explain the effect of K(+)?

Charge distribution in the active site during the reaction process was evaluated by calculating partial atomic charges with the D-RESP method. When comparing charge distributions with K⁺ vs Li⁺ present at the K1 site, we observed certain differences that seem to favour the splicing reaction in the presence of K⁺ ion (in the transition state O_{nuc} of BPA is more nucleophilic and O3'-G(-1) is a better leaving group), however they were within the range of fluctuations, making their relevance questionable. Thus, while it is possible that alterations in charge distribution do influence the reaction free energy profile, their contribution is small and cannot explain the observed difference in the free energy barrier.

To clarify this point, we have included an additional panel to Supplementary Figure 11, depicting the differences in the charge distribution in the rate-limiting transition state and added a comment in the caption to the figure.

3. Does the K(+) induce an electric field effect that assist in the reaction? Again some links to previous work may be given.

We analysed how K⁺ and Li⁺ ions influence the local electric field in the active site in the rate-limiting transition state (TS1). A figure depicting electric field forces acting on the reactive atoms has been added to Supplementary Information (Supplementary Figure 12) and discussion of results was added to the main text along with citations to previous work (page 13). The results of these additional calculations show that while in both cases the electric field forces seemed to be in-line with the movement of atoms during the splicing reaction, their orientation is more optimal in the presence of the K⁺ ion.

4. Methods: Total number of atoms in the model.

The total number of atoms has been added on page 18.

5. Methods: Give protonation states of ionizable residues.

We have now listed protonation states of ionisable residues in Supplementary Table 1. All nucleotides were in their standard protonation state.

6. Have calculations without K(+) been performed?

The calculations without K⁺ have not been performed, since crystallographic and biochemical studies on the related group II intron system showed that removing the monovalent ion effects a large disruption of the active site by breaking the triple helix required for splicing to occur. The same deformation does not seem to occur when K⁺ is substituted with Li⁺, as purified spliceosome in Li⁺-based buffers remained catalytically active.

Interestingly, it was suggested that active site disruption linked to K⁺ unbinding might also serve to push the group II intron from the first splicing step towards the second splicing step. We are currently investigating whether the same occurs also in the spliceosome complex.

We would like to thank the reviewers for his/her comments and suggestions that further improved the present works.

Reviewer #2 (Remarks to the Author):

Aupic et al. present molecular dynamics simulations of the active site of the spliceosome in the presence of the two catalytic magnesium ions as well as a highly conserved potassium ion. This potassium ion is found in the active sites of both the spliceosome and the evolutionarily related group II intron. It was hypothesized that this potassium ion played an important role in catalyzing RNA splicing, but the details were not clear. Through their molecular dynamics simulations, Aupic et al. provide a detailed model for how this potassium ion is essential for efficient RNA splicing. They found that the potassium ion plays a key role in modulating catalysis and is responsible for positioning the splice site substrates within the active site. This work is very significant in that it provides new detailed insight into the mechanism of RNA splicing at the atomic level and explains the conservation of active site metal ion architecture in both the group II intron and the spliceosome. I highly recommend publication of this manuscript in Nature Communications in its current form with no revisions required.

We thank the reviewer for his/her kind comments.

Reviewer #3 (Remarks to the Author):

Aupic and coworkers present an interesting study unraveling the overlooked contribution of monovalent ions, namely potassium, on the first transesterification reaction of the spliceosome. Although the work is completely computational, the results of the simulations are compared with experimental data. Therefore, the manuscript should be interesting for both computational and experimental readers of Nature Communications.

The manuscript is well written and the conclusions are nicely supported by the results of the

simulations, as well as by comparison with other nucleic acid processing enzymes. Nonetheless, the authors should clarify some points and add some further analyses before the manuscript can be accepted for publication, in my opinion. Please see below a detailed list of comments.

- Page 6, lines 114-118: Can the authors show the time evolution of the distance between the two Mg ions, as well as the distance between the closest Mg ion and K? How do these distances evolve (and potentially change) during the reaction?

At the start of the splicing reaction Mg²⁺ ions (M1 and M2) were located 4.1 ± 0.1 Å apart, while the M2 – K⁺ distance was 4.3 ± 0.2 Å. While the former slightly increased in the product state (4.4 ± 0.2 Å), the latter didn't show significant change. Replacement of K⁺ ion with Li⁺ did not affect the M1 – M2 distance, however the M2 – Li⁺ displayed large fluctuation in both unbiased and biased QM/MM simulations due to unstable binding of the Li⁺ ion at the K1 site.

The time evolution of the distances between Mg²⁺ and K⁺/Li⁺ ions during unbiased QM/MM MD have been included in Supplementary Figures 3 and 8. Their values at different points during the reaction have been added to Supplementary Figures 4 and 10. Short comments of their values have been added to pages 6, 11 and 13.

- Page 7, lines 134-136: Can the authors provide further details about the previous static calculations with which they compare? Could it be that the difference in free energy barrier and exothermicity of the reaction is due not only to the lack of K in the previous study, but also to the different level of theory used/neglect of system dynamics/other factors to a lower extent?

Discussion of differences in the methodological approach and obtained results between the past and the current study has been expanded on pages 9 and 16 in the main text. In short, the previous study used QM/MM nudged elastic band method, where the system, at each point on the reaction pathway, is optimized in the static regime. Additionally, the study used a starting spliceosome structure that required remodelling of the position of Mg²⁺ ions in the active site, since it was the best among available at the time. Thus, the comparison of their results to the free energy profile obtained in the presence of K⁺ ions in the current work is not perfect and the described differences in methodology could indeed have contributed to the observed discrepancies. However, the comparison to simulations with Li⁺ at the K1 confirmed the stabilizing effect of the K⁺ ion.

- Page 9, lines 167-170: Can the authors clarify whether the water molecule stabilizing the phosphorane intermediate is treated at the MM or the QM level? As far as I can tell from Figure 3I, this water molecule is not part of the coordination sphere of Mg ion, is my understanding correct?

The water molecule approached the scissile phosphate group from the bulk solvent treated at the MM level and was not part of the Mg ions coordination sphere. After observing hydrogen bonding to the intermediate we included this water in the QM zone. In our simulations, this water molecule did not participate in the proton transfer from G(+1)-O(S_p) to G(-1)-O3' (PT2) as the hydrogen bond broke beforehand. We clarified this point on page 9 of the main text.

- Page 11: As far as I understand from the Methods section, the initial snapshots for the QM/MM simulations with Li were taken from the K simulations, replacing one monovalent ion by the other. If

my understanding is correct, can the authors speculate whether further sampling of the conformational flexibility with Li using classical MD simulations could reveal further structural changes? In addition (and out of curiosity), I was wondering if there is experimental data for the substitution of K by other monovalent ions, e.g. Na.

Our simulations suggest that replacing the K⁺ ion with Li⁺ decreases the stability of the active site, resulting in increased fluctuations of reactive atoms and a less favourable reaction free energy profile. While it is possible that replacing the K⁺ ion with Li⁺ in the spliceosome system could cause further structural changes, and classical MD simulation with enhanced sampling techniques would be suitable to address this question (though performing classical MD with divalent ions is always perilous), we believe this is unlikely, since purified spliceosomes in Li⁺ buffers did not lose catalytic competence. Conversely, comparison with group II introns suggests that the absence of a monovalent ion at the K1 site might indeed lead to large structural changes in the active site and this line of research is something we are currently pursuing.

Experimental data for the substitution of K⁺ by Na⁺ is not available. In addition to Li⁺, K⁺ has been substituted by NH₄⁺, which has a similar ionic radius as K⁺. In this case, splicing efficiency was not negatively affected.

- Page 17, line 336: I believe there is a typo in the sentence "While splicing factors Cwd25 is also essential [...]"

The typo has been corrected.

- Page 17, second paragraph: Can the authors indicate the total number of atoms of the simulated system?

The total number of atoms has been added on page 18.

Is there any rationale why was the system neutralized with Na ions instead of K? Do the authors expect any difference using Na or K? or no other monovalent ions bind to the spliceosome structure during the simulations?

We agree with the reviewer that the choice of K⁺ ions for neutralization would be more suitable. However, we do not expect that exchanging Na⁺ ions with K⁺ would alter simulation results, since the choice of neutralizing ions was previously shown to not affect the simulation outcomes for nucleic acid-based systems (e.g. *Phys. Chem. B* **2012**, *116*, 9899–9916, doi: 10.1021/jp3014817, *Nucleic Acids Res.* **2006**; *34*(2), 686–696, doi: 10.1093/nar/gkj434), as well as the spliceosome in particular (*Biomolecules*, **2019**, *9*(10), 633, doi: 10.3390/biom9100633). Additionally, no additional monovalent ion binding to the spliceosome was observed in the current study.

The authors mentioned they use distance restraints between Mg ions and their coordinating atoms during the classical MD to avoid the known issues of classical force fields describing divalent ions, but why such distance restraints were also needed for K and its coordinating atoms? Did the authors want to avoid some ion/water exchange with the solution?

The reviewer is correct that classical force fields are in general suitable for description of monovalent ions. However, since the central focus of the study was the effect of the K⁺ ion on the first splicing reaction, we chose the conservative route of restraining the K⁺ ion to its binding site with harmonic bonds during classical molecular dynamics simulation to avoid its distortion. Our rationale was that

any instability at the K1 site due to errors in structure determination or ion mobility would be discovered during more rigorous unbiased QM/MM MD simulations where K^+ and its coordinating residues were treated at the QM level and no restraints were employed. The binding site remained stable during unbiased QM/MM MD and during simulations of the reaction process, indicating the K^+ ion is a structural ion. Conversely, Li^+ was not able to bind stably at the K1 site showing large positional fluctuation already during unbiased QM/MM MD, confirming the validity of our approach.

Exchange of coordinating water molecules at the K1 site was observed both in simulations with K^+ and Li^+ ion. A comment regarding this was added to the Methods section (page 19). This point is discussed also in the paragraph below.

- Page 18, QM/MM section: Can the authors clarify if there are water molecules in the first coordination sphere of any of the two Mg ions that might have been treated at the QM level? How many QM atoms are in total? I believe it would be informative to include in the SI a figure showing the QM region, as well as indicating the link atoms or capping hydrogens used to saturate the QM region, if any.

There are indeed water molecules in the first coordination sphere of both Mg^{2+} ions, 2 are coordinated to M1 and 1 to M2, and K^+ (1-3 molecules)/ Li^+ (2-5 molecules) ion. All of these, were treated at the QM level. While the water molecules bound to Mg^{2+} ions were stable, in case of K^+ and Li^+ we observed exchange of coordinated water molecules with the bulk solvent, which required the list of water molecules in the QM region to be continuously monitored and updated if needed. In total, there were 107-119 atoms in the QM region. A detailed description of the QM region has been added to the Methods section (page 19). Additionally, a figure depicting the QM region along with link atoms has been added to the Supplementary Information (Supplementary Fig. 1).

- Page 18, Free energy section: Can the authors spell out if the different simulation time for each RC (i.e. between 5-7 ps) depends on how fast the constraint force for the corresponding window converges? Also, as far as I remember, in the blue moon ensemble formalism by Ciccotti and coworkers there is a correction to be added that depends on the type of RC used. In particular, such correction should be zero for a simple distance but different from zero for a difference of distances. If my memory serves, have the authors considered such correction in their free energy profile?

The simulation times were extended to 7 ps at RC values -0.6 and 0.6, where proton transfers were observed, to obtain a well converged constraint force. To obtain the average constraint force, the final 3 ps of simulation time were used in all cases. This has now been clarified in the Methods section (pages 19-20).

The blue moon ensemble formalism indeed requires a correction term when a difference of two distances sharing a common atom is used as RC. However, the evaluation of the correction term for our system revealed that its contribution is negligible ($< 1\%$) and was therefore not included in the reported free energy profiles. We have added this information to the Methods section (page 20) and additionally included Supplementary Discussion in the Supplementary Information, where we report the form of the correction term required for the applied RC.

We thank the reviewer for careful reading of the manuscript and constructive comments that enhanced the scientific content and clarity of the manuscript.

Reviewer #1 (Remarks to the Author):

All points raised by the referees have been addressed well. Publication is recommended.

Reviewer #3 (Remarks to the Author):

I would like to recommend publication of the manuscript in its current form and thank the authors for addressing in detail all my comments.

Reviewer #1 (Remarks to the Author):

All points raised by the referees have been addressed well. Publication is recommended.

We would like to thank the reviewer for his/her help in improving the manuscript.

Reviewer #3 (Remarks to the Author):

I would like to recommend publication of the manuscript in its current form and thank the authors for addressing in detail all my comments.

We appreciate the reviewer's previous comments that contributed to strengthening the manuscript.